# Promotion Mechanism of Atrazine Removal from Soil Microbial Fuel Cells by Semiconductor Minerals

**Muyuan Tang, Yilun Sun, Xian Cao, Xiaoyi Jiang, Xintong Gao and Xianning Li ***

School of Energy and Environment, Southeast University, Nanjing 210096, China; 220210568@seu.edu.cn (X.J.); 230208066@seu.edu.cn (X.G.)

* Correspondence: lxnseu@163.com; Tel.: +86-18-795-882-590

**Abstract:** In recent years, soil microbial fuel cells (Soil-MFCs) have attracted attention due to their simultaneous electricity production and contaminant removal functions, but soil electron transfer resistance limits their contaminant removal effectiveness. To overcome the above-mentioned drawbacks, in this study, a dual-chamber Soil-MFC was constructed using atrazine (ATR) as the target contaminant, and the electrochemical performance of Soil-MFC and ATR removal were enhanced by semiconductor mineral addition. Analysis of atrazine was performed in soil using HPLC and GC-MS, and analysis of metallic minerals using XPS. Anodic microorganisms were determined using high-throughput sequencing technology. The results showed that the addition of $Fe_3O_4$ increased the maximum output voltage of the device by 2.56 times, and the degradation efficiency of atrazine in the soil to 63.35%, while the addition of $MnO_2$ increased the internal resistance of the device and affected the current output, and these changes were closely related to the ion dissolution rate of the semiconductor minerals. In addition, the addition of both minerals significantly increased the relative abundance of both Proteobacteria and Bacteroidota, and $Fe_3O_4$ simultaneously promoted the significant enrichment of Firmicutes, indicating that the semiconductor minerals significantly enhanced the enrichment of electroactive microorganisms near the anode. The structural equation modeling indicated that the semiconductor minerals achieved efficient degradation of ATR in the soil through a synergistic mechanism of metal ion leaching and microbial community structure changes. The detection of ATR and its degradation products in soil revealed that the degradation of ATR mainly included: (1) hydrolysis of atrazine by microorganisms to generate dehydroxylated atrazine (HYA); (2) reduced to diethyl atrazine (DEA) and diisopropyl atrazine (DIA) by extracellular electron reduction and re-dechlorination and hydrolysis to HYA. Semiconductor minerals make an important contribution to promoting microbial activity and extracellular electron reduction processes. The results of this study strengthen the power production and ATR removal efficiency of the Soil-MFC system and provide important theoretical support for the on-site removal of organic pollutants and the sustainable application of converting biomass energy into electricity.

**Keywords:** soil microbial fuel cell; semiconductor mineral; atrazine; electron transfer

## 1. Introduction

The organochlorine pesticide atrazine (2-chloro-4-ethylamino-6-isopropylamino-s-triazine, ATR), is widely used for broadleaf weed control on agricultural and non-agricultural land [1]. Unfortunately, atrazine is slow to passivate in soil and is capable of enriching organisms through the food chain, causing potent damage to the biological genetic structure and hormone metabolism [2,3]. Therefore, atrazine is considered an important environmental contaminant, and in particular, the potential carcinogenic effect of the s-triazine structure is of increasing concern [4].

As a moderately persistent chemical in soil, the degradation of atrazine in the environment mainly involves three processes: dechlorination, dealkylation, and deamination [5,6]. In recent years, researchers have broken away from the traditional methods of physical



adsorption and chemical oxidation, which are costly and prone to secondary contamination, and developed novel methods to treat atrazine in soil. Photocatalytic degradation is an environmentally sustainable treatment method for atrazine [7]. Zhang et al. [8] investigated the photodegradation of atrazine by natural montmorillonite clay and the phytohormone indole-3-acetic acid (IAA) under visible light irradiation. They found that when photoionized, hydrated electrons of IAA cause a reaction. These electrons then react with protons and dissolved oxygen to form hydroxyl radicals, which promote further degradation of atrazine. Plasma oxidation is also a competitive method for removing atrazine [9]. Aggelopoulos et al. [10] used the dielectric barrier discharge (DBD) method to remove atrazine from the soil. This process is carried out from a flat surface to a grid reactor under atmospheric pressure. The degradation efficiency of atrazine is a function of voltage and discharge frequency parameters. Although these methods are innovative, they still require additional energy and cannot achieve good economic benefits. It is worth noting that atrazine can be moderately transformed by spontaneous dechlorination and deamination processes under anaerobic conditions [11], which provides a good basis for anaerobic microbial electrochemical systems in soil atrazine removal.

As a bioelectrochemical technology, soil microbial fuel cells (Soil-MFC) have been widely used in the study of soil organic matter removal, which is an environmentally sustainable technology that does not require additional energy [12–14]. The excellent anaerobic region of Soil-MFC anode and the high enrichment of electroactive bacteria provide the possibility of an anaerobic reductive dechlorination process of atrazine. The Soil-MFC system constructed by Wang et al. significantly improved the degradation efficiency of atrazine through a bioelectrochemical reduction process [15]. Dominguez et al. [16] showed that 98% of atrazine in soil was effectively removed within two weeks by a modified soil-MFC system. However, all related studies showed significant differences in atrazine degradation efficiency in various regions of the soil between the cathode and anode of Soil-MFC [17], which is mainly due to the large internal resistance of the soil matrix and the fact that electroactive microorganisms can only achieve extracellular electron transfer processes at the nano- to micrometer scale so that atrazine could only be effectively degradation in the region near the anode of MFCs [18]. Therefore, how to expand the electron transfer distance of electroactive microorganisms in the soil matrix is the key to promoting the simultaneous reductive degradation of atrazine in all regions of the soil matrix.

Soil minerals are adhesive interfaces for microbial survival attachment, and natural minerals, such as metal sulfides/oxides, can act as electron shuttles that mediate extracellular electron transfer [19,20]. As Liu et al. found that iron minerals can accelerate the reduction of nitrate by ferric-reducing bacteria, and the process is mainly achieved through a mineral-mediated conduction band transfer mechanism [21]. In addition, extracellular electrons produced by electroactive microorganisms can be transferred to multivalent metal ions associated with minerals and completed as interspecies or microbe-matter electron transfer processes by free diffusion of metal ions [22]. The conduction band transfer of these minerals and the free diffusion of the associated metal ions enable electroactive microorganisms to achieve redox reactions across distances (centimeter scale) coupled with spatial separation in the soil environment [23]. Therefore, taking full advantage of the electron-mediated transfer and ion diffusion effects of natural minerals in the soil is an important means to promote the simultaneous reductive degradation of atrazine in all regions of the soil matrix. However, the contribution and mechanism of natural minerals to atrazine degradation in various regions of the soil matrix of Soil-MFC systems are not clear. The relationship among mineral valence changes, microbial community structure, and atrazine degradation under the condition of Soil-MFC electroactive microbial enrichment is also worth exploring.

Therefore, in this study, a two-chamber soil bioelectrochemical system (T-Soil-MFC) was constructed and the degradation rate of atrazine in T-Soil-MFC with different mineral additions was analyzed by liquid chromatography-mass spectrometry (LC-MS), high-

throughput sequencing and structural equation modeling (SEM) to (i) assess the promotion effect of mineral addition on atrazine degradation; (ii) reveal the response relationship among mineral valence change, microbial community structure, and atrazine degradation efficiency; and (iii) assess the contribution of mineral valence change to the degradation of atrazine and its mechanism.

## 2. Materials and Methods

### 2.1. Selection of Inoculated Sludge

The inoculated sludge was taken from an anaerobic tank of a sewage treatment plant in Nanjing, and the suspension concentration of the mixture was about 50 g/L. The sludge was retrieved and incubated in a sealed environment with regular addition of sodium acetate and stirring and then used as inoculated sludge after one week of domestication. The microbial community structure in the inoculated sludge is shown in Figure S1.

### 2.2. Preparation of Contaminated Soil

The test soil was collected from the agricultural land along the Yangtze River in Nanjing, and the top layer of 0–20 cm soil was collected, while plants, stones, and other debris were removed, and then sieved with a 2 mm aperture sieve after natural air-drying and stored at room temperature. The physical and chemical properties of the soil are shown in Table S1. Atrazine (CAS: 1912-24-9, $C_8H_{14}ClN_5$) was purchased from Aladdin. According to the mass fraction required for the test, a certain mass of ATR was weighed and mixed into the soil after dissolving in the organic solvent acetone, and stirred until well mixed, so as to obtain 100 mg/kg of ATR-contaminated soil.

### 2.3. Reactor Operation

The experimental reactor is a two-chamber Soil-MFC, which is divided into a cathode chamber and soil chamber, and the device structure is shown in Figure 1. The entire device is made of Plexiglas, with the length and width of the cathode chamber being 6 cm and the soil chamber being 10 cm × 6 cm × 6 cm. The anode is positioned vertically on the side of the soil chamber away from the cathode, at a distance of 0.5 cm from the side wall of the device. The cathode chamber and the soil chamber are separated by a cation exchange membrane (CEM). The anode material is carbon felt and the cathode is a stainless steel sheet with a thickness of 0.1 cm, and the size of them are both 6 cm × 6 cm, and the closed circuit is formed by an external resistance of 500 Ω in series with titanium wire. $MnO_2$ and $Fe_3O_4$ (recorded as Soil-MFC-$MnO_2$ and Soil-MFC-$Fe_3O_4$, respectively) were added at a mass ratio of 10:1 of contaminated soil to semiconductor minerals, while closed-circuit (Soil-MFC-control) and open-circuit (Soil-MFC-O) control groups were set up without the addition of semiconductor minerals.

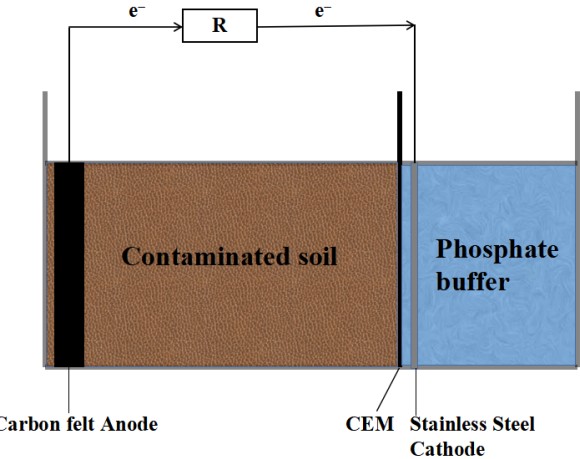

**Figure 1.** Configuration of the soil microbial fuel cell (Soil-MFC).

During the start-up phase of Soil-MFC, the anode was incubated in the domesticated treated inoculated sludge, and the anode potential was measured daily. The formal experiment was started when the anode potential stabilized. In the soil chamber, 500 g of contaminated soil and 270 mL of phosphate buffer (pH = 7) were placed, and 210 mL of phosphate buffer (pH = 2) was added to the cathode chamber. A certain amount of pure water was added to the soil chamber every 2 days during the operation of the device to keep the soil water-saturated, and 5 mL of 20 g/L sodium acetate solution was added to the anode every 7 days to maintain the anode microbial activity. The experiment was run for a total of 63 days, during which the soil was periodically (at ten-day intervals) sampled in layers according to three zones: near the anode (A), in the middle (M), and near the cathode (C). The abovementioned setup and experiment were replicated three times.

### 2.4. Electrochemical Analysis

The voltage of the closed-circuit reactor was collected by a digital acquisition system (Altech Technology Development Co., Ltd., Beijing, China), and the voltage at both ends of the cathode and anode electrodes were recorded every 30 min, and the collected data were saved automatically. The cathode and anode potential was measured with a multimeter with a saturated glycerol electrode as the reference electrode. The polarization curve was measured by a steady-state method. The soil-MFC internal resistance is the value of the slope of the polarization curve. The current value was obtained according to Ohm's law, and then the current density J and power density P were obtained according to Equations (1) and (2), respectively.

$$J = \frac{U}{R \cdot A} \tag{1}$$

$$P = \frac{U^2}{R \cdot A} \tag{2}$$

### 2.5. Pollutant and Mineral Analysis

Similar to the determination method of Wang et al. [24], 1.00 g of freeze-dried soil was weighed into a centrifuge tube, added 10 mL of methanol and 10 mL of dichloromethane, shaking for 30 min with ultrasound, and vortexing for 10 min for each sample to complete the extraction of atrazine in the soil. The extracted solution was passed through a 0.22 μm organic filter membrane and tested by high-performance liquid chromatography (HITACHI, Tokyo, Japan). The intermediate metabolites of atrazine were determined by gas chromatography–mass spectrometry (GC-MS) (Agilent, Santa Clara, CA, USA). Weigh 1.00 g of soil sample into a 50 mL centrifuge tube, add 5 mL of acetone, 5 mL of hexane, and 0.50 g of anhydrous sodium sulfate and mix uniformly, shake at room temperature for 24 h, vortex shake for 10 min, and centrifuge (3 min, 4 °C, 7000 rpm) after 30 min of ultrasonic shaking. The supernatant was filtered through 0.45 μm organic membrane, and the filtrate was transferred to a 10 mL stoppered graduated test tube, nitrogen was blown to near dryness, then dissolved in acetone and fixed to 1 mL, transferred to a small brown bottle, and refrigerated (4 °C) for measurement. The specific operating conditions for HPLC and GC-MS are described in Text S1 and S2. Weighed 1.00 g of soil samples, carbonized at 420 °C for 3 h, and digested in a microwave digester, then fixed to 10 mL with 1 mol/L nitric acid and configured with standard solutions of 0.05 to 2.0 mg/mL for the Inductive Coupled Plasma Emission Spectrometer (ICP, LEEMANLABSINC., New York, NY, USA) detection of Fe and Mn. Also, the valence changes of Mn and Fe in soil were detected using X-ray photoelectron spectroscopy (XPS, Thermo Scientific, Waltham, MA, USA) [25,26].

### 2.6. Microbiological Analysis

The anode carbon felts of each reactor were taken out at the end of the experiment, freeze-dried, and stored at −40 °C. The bacteria in the Soil-MFC anodes were sequenced and analyzed using high-throughput sequencing technology. Place the anode carbon felt in a sterile tube, add 50 mL of potassium phosphate buffer, and stir vigorously with sterile

forceps to separate the soil from the surface of the carbon felt. After removing the carbon felt, the buffer was centrifuged for 5 min, and the centrifugal precipitate was the soil containing microorganisms. The samples were weighed, put into sterile tubes, and 10 mL of 0.1 M potassium phosphate buffer was added to each gram of sample. The samples were sonicated for 1 min, vortexed for 10 s, and repeated twice, and the washing solution was removed and passed through a 0.22 μm filter membrane, and the filtered precipitate was taken for high-throughput sequencing of microorganisms.

Translated with www.DeepL.com/Translator (free version) (accessed on 2 May 2023) Genes in each sample were extracted using a soil DNA sample extraction kit and PCR amplification was performed on the V4-V5 region of the 16S rRNA. This step was repeated three times to ensure the accuracy of the experiment [27,28]. Statistical and differential analysis of the relative abundance of species was performed using SPSS (IBM SPSS Statistics 26.0) software.

### 2.7. Statistical Analysis

Statistical analysis and structural equation modeling (SEM) tests were performed for each treatment using SPSS 21.0 software [29–31]. SEM is a powerful statistical method for revealing the interactions between observed and potential variables and is widely used to explain and predict the correlation of multivariate data sets. In this study, the initial model contains electrochemical parameters (current, voltage, etc.), microbial community indices (data from NMDS Axis), mineral valence parameters (data from XPS), and pollutant degradation. A chi-square test was used to verify the quality of the fit, and the model was plausible if the $p$-value was smaller than 0.05. Correlation analysis was used to test the relationship between variables, with significance levels reported as significant (*, $0.05 > p > 0.01$) or highly significant (**, $p < 0.01$). In addition, we used the standardized total effects of SEM to identify direct and indirect relationships in the model.

## 3. Result and Discussion

### 3.1. Enhanced Electrical Performance by Mineral Leaching and Redox Conversion

Figure 2A shows the voltage changes of MFC-control, MFC-MnO$_2$, and MFC-Fe$_3$O$_4$ during system operation, with maximum output voltages of 119, 114, and 305 mV, and maximum power densities of 10.96, 3.2, and 14.13 mW/m$^2$, respectively. Compared with the control group, the maximum output voltage and power density of the MFC-Fe$_3$O$_4$ treatment group were increased by 2.56 times and 1.29 times, indicating that Fe$_3$O$_4$ had a significant promotion effect on soil-MFC power production ($p < 0.05$), while there was no significant difference in the power production performance index between MFC-MnO$_2$ and MFC-control ($p > 0.05$). From the polarization curves of the three treatment groups (Figure 2B), the internal resistance of MFC-MnO$_2$ is 8 and 5.63 times higher than that of MFC-Fe$_3$O$_4$ and MFC-control, respectively, indicating that the increased soil mass transfer resistance of MnO$_2$ was the main factor affecting the current output effect of this system, while the MFC-Fe$_3$O$_4$ treatment group had lower internal resistance relative to the control group and enhanced soil mass transfer capacity, resulting in a better electrical production performance.

In this study, we also compared the dissolved metal ions of Fe and Mn before and after the test to determine the extent of mineral involvement in redox reactions (Figure 2C). The total amount of Fe$_3$O$_4$ converted from solid to dissolved metal ions was 465.2 mg/kg, which was 9 times higher than the dissolved amount of MnO$_2$. Semiconductor minerals (Fe$_3$O$_4$ and MnO$_2$), as electron shuttles, facilitate direct electron transfer between microorganisms and electron acceptors mainly through conduction band transfer and ionic redox reactions [32,33], but conduction band transfer is accompanied by a large number of photon excitation processes so that the extracellular electron transfer process of natural minerals in the soil dark environment is mainly accomplished by the redox cycling process of ions. Both Mn and Fe elements belong to variable valence metal elements, with different valence states such as Mn$^{4+}$ and Mn$^{2+}$, Fe$^{3+}$ and Fe$^{2+}$, both can be transformed into each other through

oxidation-reduction reactions through the gain and loss of electrons [34,35]. Therefore, the massive conversion of $Fe_3O_4$ minerals to dissolved metal ions is the main reason for improving the electrical production performance of the MFC-$Fe_3O_4$ treatment group, and the interconversion of $Fe^{2+}$ and $Fe^{3+}$ realizes the electron transfer between microorganisms and anodes. In addition, Borch. et al. revealed that the soil mass transfer capacity is closely related to the metal ion content [36], and the low amount of dissolved metal ions from Mn minerals directly leads to the increase of soil mass transfer resistance and the decrease of system electrical production performance.

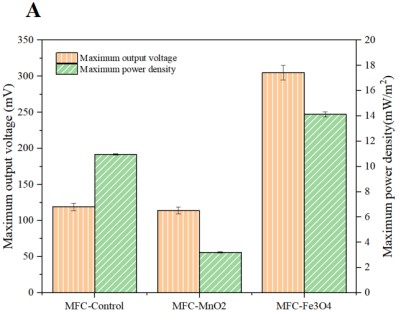
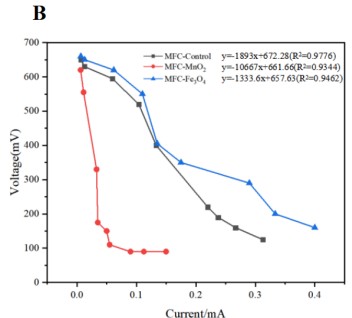
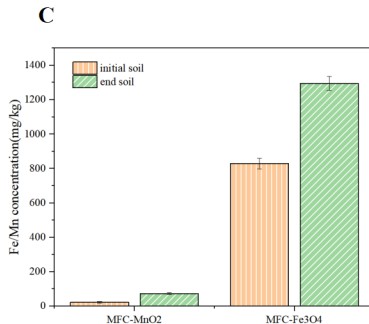

**Figure 2.** (**A**) The maximum output voltage(mV) and power density(mW/m$^2$) of MFC-Control, MFC-$MnO_2$, and MFC-$Fe_3O_4$. (**B**) The polarization curves of MFC-Control, MFC-$MnO_2$, and MFC-$Fe_3O_4$. (**C**) The dissolved metal ion of Fe and Mn before and after the test.

### 3.2. Effect of Minerals on Atrazine (ATR) Removal from Soil

The overall ATR residue concentration of soil and the ATR residue concentration at different locations of Soil-MFC were measured at the end of the experiment, to investigate the impact of different types of semiconductor minerals on ATR removal by Soil-MFC. Figure 3A shows the overall changes in ATR residues in the treatment groups. The final ATR residue concentrations of MFC-Control, MFC-$MnO_2$, MFC-$Fe_3O_4$, and MFC-O were 48.07 mg/kg, 42.65 mg/kg, 36.65 mg/kg, 60.48 mg/kg, respectively, and the removal rates showed a trend of MFC-$Fe_3O_4$ (63.35%) > MFC-$MnO_2$ (57.35%) > MFC-Control (51.93%) > MFC-O (39.52%). These results suggest that the addition of semiconductor minerals significantly improves the degradation efficiency of atrazine. Notably, although the addition of $MnO_2$ improved the degradation efficiency of ATR, the promotion effect was significantly different from that of MFC-$Fe_3O_4$ ($p < 0.05$), indicating that the number of metal ions dissolved not only affects the system current output but also has a significant effect on the extracellular electron transfer process between microorganisms and pollutants.

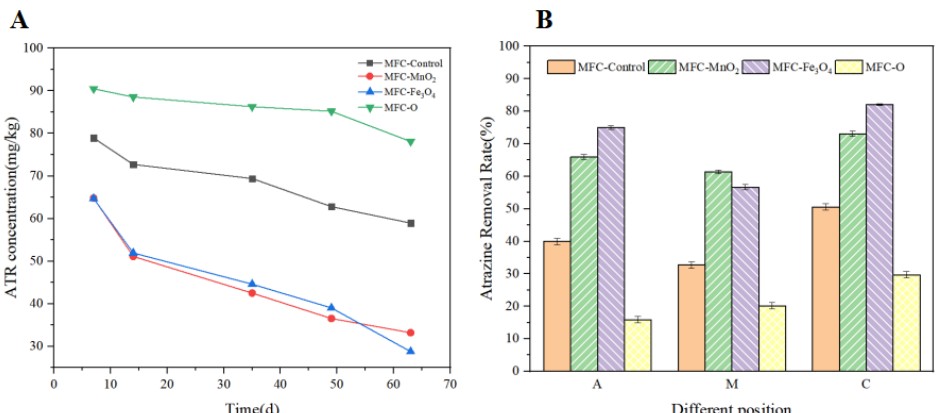

**Figure 3.** (**A**) The overall residual amount of ATR during operation in the 4 groups. (**B**) The ATR removal rate of different positions of 4 groups.

The ATR removal rate of different positions of Soil-MFC (near anode A, middle M, and near cathode C) at the end of the experiment is shown in Figure 3B. The over-

all pattern of the spatial distribution of ATR content among different treatment groups was MFC-Fe$_3$O$_4$ > MFC-MnO$_2$ > MFC-Control > MFC-O, which was the same trend as the overall ATR residual concentration of the soil after mixing. Compared with the control group, the degradation efficiency of MFC-Fe$_3$O$_4$ and MFC-MnO$_2$ in the M region was enhanced by 23.95% and 38.65%, respectively. This region was far away from the electrode, and their increased degradation efficiency was mainly due to the addition of semiconductor minerals, indicating that the dissolution of metal ions expanded the range of extracellular electron transfer of microorganisms and improved the probability of atrazine contacting electrons.

In the same treatment group, it can be observed that the lowest degradation rate appeared in the M region, and the removal rate in the C region was higher than that in the A region. The higher degradation efficiency of ATR in the C region was attributed to its presence mainly in the form of cations in the cathodic acidic environment, which could diffuse into the cathodic solution through the cation exchange membrane [16]. Besides, the effect of the electric field inside the Soil-MFC further promoted the migration of ATR ions, and at the end of the test, the presence of ATR was also clearly detected in the cathodic solution. Similarly, it has been shown that under anaerobic conditions with low cathodic potential, ATR can gain electrons in the cathodic region and achieve rapid reductive dechlorination [16]. These factors together influence the concentration of ATR in the region near the cathode.

Influenced by the saturated water conditions during the experiment, the metal ions dissolved from the minerals could be uniformly distributed in the soil matrix. However, the degradation efficiency of ATR at different spatial sites of Soil-MFC differed significantly. It suggested that the dissolution of semiconductor mineral ions was not the only factor that affected the ATR removal efficiency, the change in the microbial community structure of the soil matrix was also an important factor promoting ATR degradation. The removal of ATR in region A was associated with the enrichment of electroactive microorganisms near the anode and the increase of microbial metabolic activity. Wang et al. showed that the bioelectrochemical reactions associated with the anodic electrogenic bacteria in the Soil-MFC are the main mechanism of ATR degradation in this region [27]. Therefore, while semiconductor minerals expand the range of microbial extracellular electron transfer and promote ATR degradation in the far-anode region, they also affect the efficiency of ATR degradation by altering the microbial community structure.

### 3.3. Effects of Mineral Additions on the Microbial Community of Soil-MFC Anode

In this study, the microorganisms at the anode of each treatment group at the end of the experiment were analyzed to investigate the effect of semiconductor mineral addition on the microbial community in the system. The Coverage values of microorganisms in all treatment groups were around 0.99, indicating that the sequencing results fully reflected the composition of the microbial community structure near the anode of the device (Table S2). Table 1 shows the Alpha diversity, Ace, and Chao indices of the anode microbial community in each group. The order of microbial abundance in the four treatment groups was: MFC-Fe$_3$O$_4$ ≈ MFC-MnO$_2$ > MFC-O > MFC-Control, and the order of microbial diversity was: MFC-Fe$_3$O$_4$ ≈ MFC-MnO$_2$ > MFC-Control > MFC-O. From the microbial phylum structure (Figure 4A), the relative abundance of proteobacteria in each group is the highest, while other bacteria with relatively rich distribution mainly include Bacteroidota and Firmicutes. Proteobacteria and Bacteroidota are common microbial communities in electrochemical systems [37], which increased significantly ($p < 0.05$) with the addition of semiconductor minerals. Notably, the addition of Fe$_3$O$_4$ significantly promoted the growth of the Firmicutes, which contains lots of typical electroactive microorganisms in soils [38,39], which may be an important reason for the superior electricity production of MFC-Fe$_3$O$_4$ compared to the other groups. The above results suggest that the addition of semiconductor minerals significantly increased the abundance and structural diversity of the microbial community in the Soil-MFC anode, especially Fe$_3$O$_4$, which may promote

the enrichment of electroactive microorganisms in the soil and improve the electricity production effect of Soil-MFC.

**Table 1.** The Alpha diversity of MFC-O, MFC-Control, MFC- $Fe_3O_4$, and MFC-$MnO_2$.

| Sample | Sobs | Shannon | Simpson | Ace | Chao | Coverage |
|---|---|---|---|---|---|---|
| MFC-O | 753 | 3.02 | 0.17 | 1399.04 | 1134.49 | 0.991 |
| MFC-Control | 686 | 3.33 | 0.12 | 1271.78 | 1079 | 0.992 |
| MFC-$Fe_3O_4$ | 1115 | 4.89 | 0.03 | 1462.35 | 1482.64 | 0.989 |
| MFC-$MnO_2$ | 1092 | 4.52 | 0.04 | 1484.07 | 1479.67 | 0.989 |

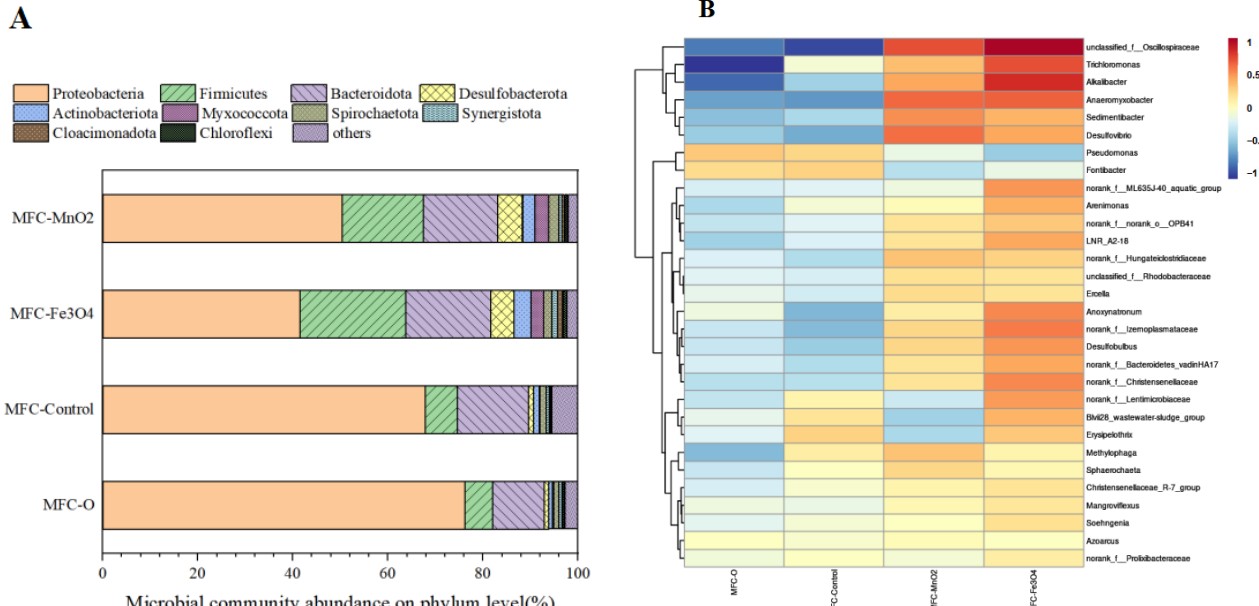

**Figure 4.** (**A**) The microbial community abundance on phylum level of MFC-O, MFC-Control, MFC-$Fe_3O_4$, and MFC-$MnO_2$. (**B**) The relative abundance of dominant microorganisms at the genus level of MFC-O, MFC-Control, MFC- $Fe_3O_4$, and MFC-$MnO_2$.

In order to assess the effect of semiconductor minerals on microbial community structure more clearly, the relative abundance of dominant microorganisms at the genus level (Top 30) was analyzed in this study, and the results are shown in Figure 4B. The addition of semiconductor minerals significantly increased the relative abundance of Anaeromyxobacter and Desulfovibrio compared to the control group, both of which are typical electroactive microorganisms. In addition, the relative abundance of Trichloromonas was also significantly increased, and some studies have shown that it can be used as the main electrogenic microorganism in electrochemical anaerobic digestion [40]. These results confirm that the mineral addition, especially $Fe_3O_4$, can significantly promote the enrichment of anode electroactive bacteria [41,42]. Anoxynatronum increased significantly in the MFC-$Fe_3O_4$ treatment group and was reported to degrade multiple carbon sources through anaerobic oxidation, which has a certain role in ATR removal and conversion [43]. Compared to MFC-$Fe_3O_4$, the relative abundance of Methylophaga was higher in MFC-$MnO_2$, which is a methanogenic bacterium that can anaerobically oxidize organic substrates to produce methane, and it may compete with electroactive microorganisms for carbon sources in the anode region, thus reducing electron production [44]. The large enrichment of this genus may be the main reason for the lower relative abundance of electroactive microorganisms in the MFC-$MnO_2$ treatment group.

### 3.4. Key Factors Affecting ATR Degradation

The abovementioned results indicated that metal ion concentration and microbial community structure had a significant impact on ATR degradation, and this study used MFC-Fe$_3$O$_4$ as an example for its extraordinary ATR degradation ability to explore the key factors affecting ATR degradation through structural equation modeling (SEM). The initial model contained six main factors (Figure 5A), including microbial community index (data from NMDS Axis), ATR degradation rate, and Fe$^{2+}$ and Fe$^{3+}$ content (data from XPS fit). The quality control parameters of the structural equation model are shown in Table S3. The latent variables Fe$^{2+}$ and Fe$^{3+}$ content showed significant differences in the effect of microorganisms, and microbial community 2 (NMDS2) had a significant positive impact on Fe$^{2+}$ ($\lambda = 0.63$, $p < 0.001$). It indicates that microbial community 2 may be the main factor affecting the dissolution and release of minerals from the soil. In addition, Fe$^{2+}$ directly and positively influenced the degradation rate of ATR ($\lambda = 0.15$, $p < 0.01$), confirming that Fe$^{2+}$ is the key factor in mediating extracellular electron transfer and promoting ATR degradation. It is noteworthy that microbial community structure 2 directly affected the ATR degradation along with Fe$^{2+}$ concentration ($\lambda = 0.22$, $p < 0.01$), indicating that microbial community structure 2 contains a large number of electron-producing microorganisms and ATR-degrading bacteria. Although the changes in Fe$^{3+}$ content did not affect the ATR degradation, it had a significant positive effect on microbial community 2.

**A**            **B**

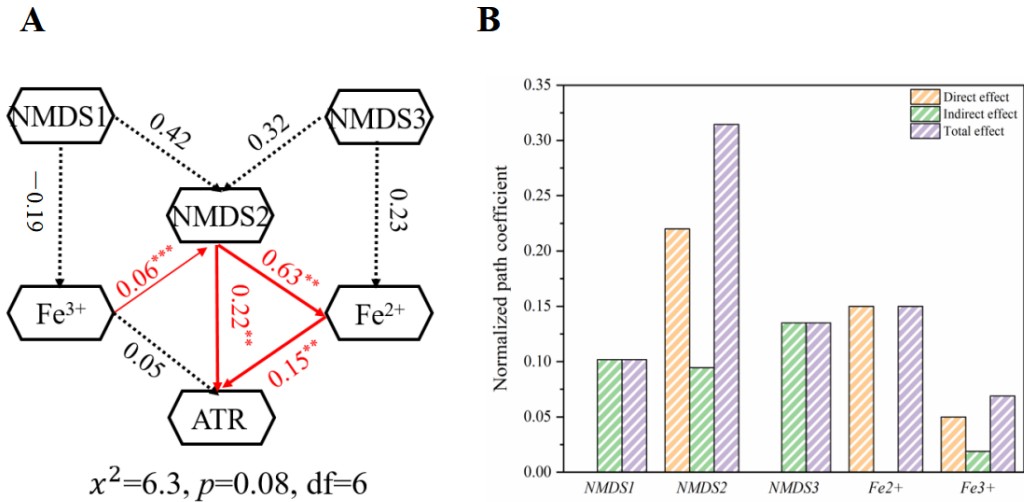

**Figure 5.** (**A**) The structural equation modeling (SEM) of the key factors affecting ATR degradation. (**B**) The standardized total effect of SEM on NMDS1, NMDS2, NMDS3, Fe$^{2+}$, and Fe$^{3+}$ (** $p < 0.01$, *** $p < 0.001$).

SEM not only demonstrated the key factors affecting ATR degradation but also the contribution of Fe$^{2+}$ dissolution and microbial contribution to ATR degradation can be seen from the standardized total effect of SEM (Figure 5B). The results of the standardized total effect showed that the dissolution process of Fe$^{2+}$ had a direct contribution of about 15% to ATR degradation, while the direct contribution of microbial community structure 2 to ATR degradation was as high as 22%. In addition, the indirect contribution of microbial community structure 2 to ATR degradation was as high as 9.5%, indicating that ATR degradation is closely related to microbial effects and the extracellular electron transfer process, and the extracellular electron transfer process makes a non-negligible contribution to ATR degradation. In conclusion, the SEM results confirmed that semiconductor minerals achieved efficient ATR degradation in soil under the dual mechanism of microbial effect and Fe$^{2+}$ solubilization.

### 3.5. Degradation Mechanism of ATR in Soil-MFC

In this study, ATR and related degradation products in the soil substrates of soil-MFC were examined to determine the degradation process of ATR in Soil-MFC. ATR residues and their three degradation products, deethylatrazine (DEA), deisopropylatrazine (DIA), and hydrocarbon-based atrazine (HYA), were detected in both treatment groups with/without the addition of semiconductor minerals. Among them, DEA and DIA are the main chlorinated products of ATR after dealkylation, and HYA is the hydrolysis product of ATR. Therefore, there are two main degradation pathways of ATR under the operation of Soil-MFC: (1) hydrolysis to generate HYA-like substances under the effect of microorganisms; and (2) reduction of DEA and DIA through extracellular electron transfer, and then conversion to HYA through dechlorination hydrolase or pH influence. During subsequent degradation, the HYA generated by those two paths can be deaminated to form cyanuric amide, which is then converted to cyanuric acid, and ultimately decomposed into $CO_2$ and ammonia under the action of amidohydrolase, etc. (Figure 6).

**Figure 6.** Degradation pathways of ATR in soil.

From the abovementioned results, it can be seen that the degradation process of ATR involves a large number of reduction reactions that can be influenced by extracellular electron transfer in Soil-MFC. Therefore, the addition of semiconductor minerals can effectively promote the conversion of ATR to substances such as DEA and DIA while increasing the electron transfer rate in the system, and the rapid formation and degradation of DEA and DIA may be the main reason for the accelerated ATR mineralization process in the electrochemical system. The structural equation model also indicates that mineral addition contributes about 15% to the degradation of ATR. In addition, although the degradation of ATR involves a large number of reduction reactions, the overall mineralization process is still closely related to the biological activities of some functional microorganisms, such as most degrading bacteria promote the conversion of ATR to HYA and DIA through the secretion of various hydrolytic enzymes (ethylamine hydrolase, chlorine hydrolase). The above processes are very slow under natural conditions, but in bioelectrochemical systems, especially after the addition of semiconductor minerals, the evolution of the microbial community strengthens the biodegradation pathway of ATR. Eventually, ATR is efficiently degraded in Soil-MFC by biotic/abiotic pathways.

Importantly, only degradation products such as DEA, DIA, and HYA were detected in the Soil-MFC matrix in this study, indicating that the ATR in Soil-MFC was only initially degraded at about 60 d of device operation, and a much longer process is required to achieve complete mineralization of ATR. Therefore, in order to further improve the degradation

of ATR by Soil-MFC and related microorganisms, it is necessary to continue to enhance the extracellular electron transfer process in the soil to break through the key rate-limiting step of ATR degradation in the soil environment, and to increase the degradation enzyme activity to accelerate the mineralization process of intermediate metabolites.

*3.6. Environmental Impact*

Atrazine is widely used as a herbicide in agricultural production due to human activities, but it poses a significant risk to human health for its hard-to-degrade nature. Traditional remediation methods such as physical adsorption, immobilization, and chemical oxidation have obvious limitations and may pose new environmental risks. Remediation of atrazine using a combination of microbial and electrochemical methods can not only remove contaminants but also bring new energy sources. However, microbial electrochemical methods when applied to the soil are limited by the weak mass transfer properties of the soil, and the extracellular electron transfer of electroactive microorganisms in the soil is limited to the anode region. Therefore, in this study, the natural minerals $Fe_3O_4$ and $MnO_2$ were applied as electron shuttles to mediate electron transfer and promote the remediation of atrazine in soil. The experimental results showed that the application of $Fe_3O_4$ could significantly promote the enhancement of electron production and atrazine removal, which implies that the addition of minerals to the soil MFC system for atrazine degradation in the soil is an effective technique at the laboratory scale.

This study was able to enhance atrazine removal from the reactors in small-scale experiments, but the application of these reactors at the engineering level has not been reported. Scaling up these reactors for in situ treatment of contaminants in real-field scenarios is a promising area of research. Microbial fuel cells not only contribute to the degradation of atrazine but also allow for bioelectricity production. In addition, there are many uncharted areas in the mechanism of mineral-facilitated electron transfer in soils, such as how changes in mineral structure during the reaction process affect electron transfer and how it affects contaminant removal.

**4. Conclusions**

Low electron transport efficiency is a key factor limiting the removal of soil organic contaminants by bioelectrochemical systems. In this study, natural minerals were used as electron shuttles to effectively enhance the high electron transport efficiency of soil-MFC and promote electricity production and ATR degradation. The rate of metal ion leaching is the key factor in determining electron transport efficiency, and SEM directly showed that $Fe^{2+}$ leaching has a direct contribution of about 15% to ATR degradation. The addition of natural minerals simultaneously changed the microbial community structure in the anode region of Soil-MFC and influenced the ATR degradation. In conclusion, natural minerals synergistically contributed to the removal of ATR from the soil through both ion solubilization and modification of microbial community structure. This study provides theoretical support to further improve the removal of environmental organic pollutants by Soil-MFC.

**Supplementary Materials:** The following supporting information can be downloaded at: https:// www.mdpi.com/article/10.3390/su15097706/s1, Table S1: Basic physical and chemical properties of original soil; Table S2: The coverage index of MFC-O, MFC-Control, MFC-$Fe_3O_4$, MFC-$MnO_2$; Table S3: Quality control parameters of Structural equation models; Figure S1: Distribution of dominant microorganisms in original soil at phylum and genus levels.

**Author Contributions:** Software, X.J. and X.G.; Data curation, Y.S.; Writing—original draft, M.T.; Writing—review & editing, X.C. and X.L. All authors have read and agreed to the published version of the manuscript.

**Funding:** This work was financially supported by the National Natural Science Foundation of China (42077108).

**Institutional Review Board Statement:** Not applicable.

**Informed Consent Statement:** Not applicable.

**Data Availability Statement:** No new data were created.

**Conflicts of Interest:** The authors declare no conflict of interest.

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
