# Peer review of "Promotion Mechanism of Atrazine Removal from Soil Microbial Fuel Cells by Semiconductor Minerals"

_sustainability, doi:10.3390/su15097706_

Round 1
Reviewer 1 Report
Please, kindly find the comments below:
1. Abstract should briefly give a background to the current study. What existing problem are the authors trying to solve, and what are the gaps in the existing literature they seek to fill
2. ATR should be stated fully in its first appearance in the abstract.
3. In the methodology, values of some experimental conditions such temperature, time, volume, mass should be justified. Were those values selected at random or based on previous knowledge of how it affect results or based on the physicochemical characteristics of the samples under investigation? These reasons should be evident in the methodology to give a basis as to why those values were chosen for the experiment. For example, “..ultrasonic shaking (3 min, 4oC, 6000 rpm)”, why not a higher shaking time, higher or lower stirring temperature? That’s why it is important to validate or explain your selected experimental values.
4. In experimental setups like the current study, the experiment is repeated at least three times and the average of the three trials is selected as the final value. The results for figures are then plotted with error bars. The authors mention that their experiment was replicated thrice but I see nothing of that sort in their figures. Why is this so? I think it is important to include error bars to indicate the range of all possible values and also how much the individual trials deviated from one another.
5. Current conclusion could be improved. Reiterate what the existing problem was and what the current study attempted to solve the problem. The current findings should be summarized in a way that captures all the key results in the study. Recommendations for future works to develop what has currently been done needs to be included if any.
Author Response
Point 1: Abstract should briefly give a background to the current study. What existing problem are the authors trying to solve, and what are the gaps in the existing literature they seek to fill.
Response 1: We have improved the Abstract section by adding a brief background of the study on soil microbial fuel cells and pointed out that the main problem we were trying to solve is the low organic pollutants removal rate caused by the low electron transfer in the soil. This is a part that is rarely addressed in the existing literature and we highlight the role of adding minerals as electron mediators in the soil-MFC system(line 8-13).
Point 2: ATR should be stated fully in its first appearance in the abstract.
Response 2: We have improved this problem in the abstract, i.e. ATR first appeared in full as atrazine(line 11).
Point 3: In the methodology, values of some experimental conditions such temperature, time, volume, mass should be justified. Were those values selected at random or based on previous knowledge of how it affect results or based on the physicochemical characteristics of the samples under investigation? These reasons should be evident in the methodology to give a basis as to why those values were chosen for the experiment. For example, “..ultrasonic shaking (3 min, 4oC, 6000 rpm)”, why not a higher shaking time, higher or lower stirring temperature? That’s why it is important to validate or explain your selected experimental values.
Response 3: In this study, we chose the experimental conditions based on following reasons:
- For the selection of temperature: In our system, microorganisms were the main factor causing power production and contaminant removal, so we chose a temperature of 20-30°C that was optimal for mixed bacterial growth.
- For the selection of time: The current output in our system remained stable throughout the experimental run time and the degradation of ATR reached the desired state within 60 days, so we chose to stop the experiment at 60 days.
- For the selection of volume and mass of soil: The volume and mass of the soil depends on the size of the device, which depends on the size of the electrodes and distance between them. We have analyzed previous studies, as shown in the literature Effects of cathode/anode electron accumulation on soil microbial fuel cell power generation and heavy metal removal(doi: doi.org/10.1016/j.envres.2021.111217), we have chosen the electrodes size as well as the distance that were the most conducive to achieve our goals.
- For the selection of atrazine extraction conditions: Atrazine extraction conditions have been developed among previous studies with well-established methods, as presented in the literature Detection of nine herbicide residues in Angelica sinensis by QuEChERS-GC-MS(doi: 10.7671/j.issn.1001-411X.2017.06.012), Residual Detection of Triazine Herbicides in White Wine Using SPME - GC /MS(doi: 10.14083/j.issn.1001-4942.2018.11.026.) We refer to the previous methods for the extraction of atrazine to ensure the accuracy of the experiment.
Point 4: In experimental setups like the current study, the experiment is repeated at least three times and the average of the three trials is selected as the final value. The results for figures are then plotted with error bars. The authors mention that their experiment was replicated thrice but I see nothing of that sort in their figures. Why is this so? I think it is important to include error bars to indicate the range of all possible values and also how much the individual trials deviated from one another.
Response 4: We’re sorry that we made the mistakes in the drawing. We value the error nature of the experiments and have redrawn the plots, and made error bars for all parallel trials(line 233).
Point 5: Current conclusion could be improved. Reiterate what the existing problem was and what the current study attempted to solve the problem. The current findings should be summarized in a way that captures all the key results in the study. Recommendations for future works to develop what has currently been done needs to be included if any.
Response 5: We have improved the Conclusion section by reiterating the existing problems and how to improve them. In addition, we have added an Environmental Impact section to emphasize the direction of future work and the significance of this study for real-field applications(line 444-455).
Reviewer 2 Report
Kindly find the below comments.
Abstract:
-Define ATR in abstract.
-The purpose and significance of the study should be clearly stated in the abstract. What problem does this research aim to solve? What is the potential impact of the findings?
-The methodology of the study should be briefly explained in the abstract.
-The key results of the study should be clearly summarized in the abstract.
-The implications of the study should be discussed in the abstract. How do the findings advance our understanding of atrazine removal and microbial fuel cell technology? What are the practical applications of the research?
Introduction:
The introduction is well-written and provides a clear and concise overview of the topic being discussed. The author has done an excellent job of setting the stage for the rest of the paper and has provided the reader with a good understanding of the importance of the subject matter.
-Kindly provide a photo of the experimental setup.
-Please mention teh size of electrodes and membrane
-Section 2.6: Please exlplain how the biofilm were seperated from the electrode for DNA extraction?
-Please clearly explain how internal resistance is measured?
-3.1: the text doesnt match with values, please chcek and fix:
"Figure 1A shows the voltage changes of MFC-control, MFC-Fe3O4, and MFC-MnO2 during system operation, with maximum output voltages of 119, 114, and 305 mV, and maximum power densities of 10.96, 3.2, and 14.13 mW/m2 , respectively. Compared with the control group, the maximum output voltage and power density of the MFC-Fe3O4 treatment group were increased by 2.56 times and 1.29 times, indicating that Fe3O4 had a significant promotion effect on soil-MFC power production (p<0.05), while there was no significant difference in the power production performance index between MFC-MnO2 and MFC-control (p>0.05)."
- Justifications and interpretation of the data with the literature in the results and discussion sections should be improved.
Conclusion:
-The conclusion could be further strengthened by providing more specific information about the results, such as the magnitude of the effects observed with the addition of minerals and metal ions.
-The authors could also consider discussing the potential practical applications of their findings.
-It would be helpful to include some limitations of the study and suggestions for future research. For example, the authors could discuss potential factors that may affect the reproducibility of the results, or identify areas for further investigation to better understand the mechanisms underlying the observed effects.
-
Author Response
Point 1: Define ATR in abstract.
Response 1:We have improved this problem in the abstract, i.e. ATR first appeared in full as atrazine(line 11).
Point 2: The purpose and significance of the study should be clearly stated in the abstract. What problem does this research aim to solve? What is the potential impact of the findings?
Response 2: We have improved the Abstract section by adding a brief background of the study on soil microbial fuel cells and pointed out that the main problem we were trying to solve is the low organic pollutants removal rate caused by the low electron transfer in the soil. This is a part that is rarely addressed in the existing literature and we highlight the role of adding minerals as electron mediators in the soil-MFC system(line8-13).
Point 3: The methodology of the study should be briefly explained in the abstract.
Response 3: We improved this problem by adding the main study methodology to the Abstract section(line 13-14).
Point 4: The key results of the study should be clearly summarized in the abstract.
Response 4: We have improved the Abstract section by spelling out the main findings(line 15-29).
Point 5: The implications of the study should be discussed in the abstract. How do the findings advance our understanding of atrazine removal and microbial fuel cell technology? What are the practical applications of the research?
Response 5: We have improved the Abstract section to describe the significance of the choice of atrazine removal and the significance of microbial fuel cell technology for power production and contaminant removal. In addition, we have added the significance of this study for practical applications, which mainly include providing important theoretical support for the on-site removal of organic pollutants(line 8-13, 29-32).
Point 6: Kindly provide a photo of the experimental setup.
Response 6: We placed the experimental setup schematic in the supplementary material in our first draft, and we have now moved it to 2.3 section in the main text to show our experimental setup more clearly(line 133-134).
Point 7: Please mention teh size of electrodes and membrane.
Response 7: We have added the size of electrodes and membrane in the 2.3 section(line 127).
Point 8: Section 2.6: Please exlplain how the biofilm were seperated from the electrode for DNA extraction?
Response 8: We have added this part in the 2.5 section. Specifically, we weigh 1.00 g of soil sample into a 50 ml centrifuge tube, add 5 mL of acetone, 5 mL of hexane and 0.50 g of anhydrous sodium sulfate and mix uniformly, shake at room temperature for 24 h, vortex shake for 10 min, and centrifuge (3 min, 4 ℃, 7000 rpm) after 30 min of ultrasonic shaking. The supernatant was filtered through 0.45 μm or-ganic membrane, and the filtrate was transferred to a 10 mL stoppered graduated test tube, nitrogen blown to near dryness, then dissolved in acetone and fixed to 1 mL, transferred to a small brown bottle, and refrigerated (4 ℃) for measurement.
Point 9: Please clearly explain how internal resistance is measured?
Response 9: We have added this part in the 2.4 section. Specifically, the soil-MFC internal resistance is the value of the slope of the polarization curve, and the polarization curve was measured by a steady-state method.After the MFC was disconnected for 12 h, the resistance of the external resistor was changed from 100000 Ω to 50 Ω, and measure the voltage after the device has been stabilized for 1 h. In the ohmic loss region, the current and current density are approximately linear. Therefore, the internal resistance can be calculated based on the linear fit in this region.
Point 10: 3.1: the text doesnt match with values, please chcek and fix:
"Figure 1A shows the voltage changes of MFC-control, MFC-Fe3O4, and MFC-MnO2 during system operation, with maximum output voltages of 119, 114, and 305 mV, and maximum power densities of 10.96, 3.2, and 14.13 mW/m2 , respectively. Compared with the control group, the maximum output voltage and power density of the MFC-Fe3O4 treatment group were increased by 2.56 times and 1.29 times, indicating that Fe3O4 had a significant promotion effect on soil-MFC power production (p<0.05), while there was no significant difference in the power production performance index between MFC-MnO2 and MFC-control (p>0.05)."
Response 10: We are sorry we made a mistake in the text description, now we have fixed this bug. Thank you for pointing that out(line 209).
Point 11: Justifications and interpretation of the data with the literature in the results and discussion sections should be improved.
Responese 11: We have improved Results and Discussion section. We referred to more recent literature for further elucidation of the relationship between metallic minerals and electricity production and contaminant removal.
Point 12: The conclusion could be further strengthened by providing more specific information about the results, such as the magnitude of the effects observed with the addition of minerals and metal ions.
Response 12: We have improved the Conclusion section by providing more specific information about the result, reiterating the existing problems and how to improve them. In addition, we emphasize the significance of this study for real-field applications(line 453-455).
Point 13: The authors could also consider discussing the potential practical applications of their findings. It would be helpful to include some limitations of the study and suggestions for future research. For example, the authors could discuss potential factors that may affect the reproducibility of the results, or identify areas for further investigation to better understand the mechanisms underlying the observed effects.
Response 13: We have improved these problems by adding the potential practical applications in the Conclusion section, and we have added an Environmental Impact section to emphasize the direction of future work(line 420-442).
Reviewer 3 Report
The present manuscript entitled “Promotion mechanism of atrazine removal from soil microbial fuel cells by semiconductor minerals” have the authors studied to a dual-chamber soil microbial fuel cell (Soil-MFC) was evaluated to the promotion effect of semiconductor mineral addition on atrazine removal and analyzed the key factors affecting ATR degradation. The results showed that the addition of Fe3O4 increased the maximum output voltage of the device by 2.56 times, and the degradation efficiency of atrazine in the soil to 63.35%, while the addition of assessed the promotion effect of mineral addition on atrazine degradation and reveal the response relationship among mineral valence change, microbial community structure and atrazine degradation efficiency, and its mechanism.
The reported work obtained results is encouraging and more precisely fits into the journal. There are some queries to be taken up, so that the manuscript will be clearer to the readers.
1. Introduction can be change, it is looks like somewhat monotonous; it can improve use to recent literature.
2. Clearly improve the quality of the Figures.
3. Authors must define clearly the novelty and the critical improvements in your paper compared to other similar works.
4. Rewrite the conclusion section, for more clearly mention the novelty of the paper.
English language refinement will improve the readability of the manuscript.
Author Response
Point 1: Introduction can be change, it is looks like somewhat monotonous; it can improve use to recent literature.
Response 1: We have improved the Introduction section by using more recent literature. Progress in atrazine removal from soils is recapitulated to provide updated and useful information for this section(line 47-55).
Point 2: Clearly improve the quality of the Figures.
Response 2: We have redrawn the image to ensure its accuracy and clarity.Thank you for pointing that out(line 223, 272).
Point 3: Authors must define clearly the novelty and the critical improvements in your paper compared to other similar works.
Response 3: We have improved the Abstract, Introduction, Discussion and Result and Conclusion section, to emphasize what makes our study different from other similar studies.Importantly, the application of metallic minerals as electron mediators in soil microbial fuel cells for theoretical studies is something that has been little studied before.
Point 4: Rewrite the conclusion section, for more clearly mention the novelty of the paper.
Response 4: We have rewritten the results section to reformulate the problem we tried to solve and to recapitulate the main conclusions we reached, with emphasis on the effect of metal ion solubilization on ATR degradation, which is a novel part of our study(line 444-455).
Round 2
Reviewer 1 Report
The article is modified accordingly.
Author Response
Dear reviewer,
Thank you for your response sincerely!